# OpenReview forum: "ASIDE: Architectural Separation of Instructions and Data in Language Models"
_ICLR.cc/2026/Conference — ICLR 2026 Poster_

### Official Review · Reviewer_Rtoh · 2025-10-27

**Soundness:** 4
**Presentation:** 4
**Contribution:** 4
**Rating:** 8
**Confidence:** 4

**Summary:**

This paper presents a new method to defend against prompt injection attack by separating instructions and data at the level of token embeddings through orthogonal rotation. The advantage of the proposed method is that it does not rely on safety-related data for training while achieving good performance on both normal tasks and safety tasks.

**Strengths:**

1. This method is novel to my knowledge.

2. The safety performance of the proposed method is promising.

3. The proposed method can achieve better safety without the need of safety data.

4. Many different benchmark datasets are used for evaluation.

**Weaknesses:**

1. It is not very clear on the selection of orthogonal rotation. Why is it better than other kinds of transformation? Is there any theoretical analysis on this?

2. Orthogonal rotation is simple, which is good. But does it can well fit different data distributions, tasks and models?

3. It would be better if more normal tasks such as reasoning-related tasks are incorporated into experiments to evaluate the impact of the proposed method on model performance.

**Questions:**

Please refer to above comments.

---

> ### Author Response · Authors · 2025-11-20
>
> Dear Reviewer Rtoh,
>
> We thank you for your review and your appreciation of the proposed method. We are pleased to address the points you raised below:
>
> > It is not very clear on the selection of orthogonal rotation. Why is it better than other kinds of transformation? Is there any theoretical analysis on this?
>
> Theoretically, orthogonality ($\theta = \pi/2$) enforces zero correlation between the instruction and data subspaces, maximizing their distinguishability.
>
> To demonstrate this empirically, we ran an additional ablation study training ASIDE for Qwen3-8B with different rotation angles$\theta \in \{\pi/8, \pi/4, 3\pi/8, \pi/2, 5\pi/8, 3\pi/4, 7\pi/8, \pi\}$. We observed a clear trend: our separation metric (SEP) increases monotonically as $\theta$ moves from $0$ to $\pi/2$ (90 degrees), and then decreases for larger angles. We have added this analysis to **Appendix A.1**.
>
> > Orthogonal rotation is simple, which is good. But does it can well fit different data distributions, tasks and models?
>
> We devised ASIDE as a model-agnostic method for enhancing instruction-data separation. It is specifically designed for integrated applications (e.g., email clients) or safety scenarios with clear instruction-data distinctions, such as black-box monitoring. Thus, ASIDE adapts well to any task within this category, remaining agnostic to the exact data distribution or model architecture.
>
> > It would be better if more normal tasks such as reasoning-related tasks are incorporated into experiments to evaluate the impact of the proposed method on model performance.
>
> We agree that adding additional tasks would be nice. Unfortunately, there’s no good training dataset with reasoning traces that has instruction-data labels. We follow all other works in the field that are training on Alpaca, e.g., ISE([1], ICLR’25), and so do recently published works, e.g., CAHL([2], NeurIPS’25) and FocalLoRA([3], NeurIPS’25).
>
> - - -
>
>
> Once again, we thank you for your strong assessment of our work. We hope that we have addressed your concerns, and we would be happy to answer any further questions.
>
> - - -
>
> References:
>
> [1] Wu, T., Zhang, S., Song, K., Xu, S., Zhao, S., Agrawal, R., Indurthi, S. R., Xiang, C., Mittal, P., & Zhou, W. (2025). Instructional Segment Embedding: Improving LLM Safety with Instruction Hierarchy. International Conference on Learning Representations (ICLR 2025).
>
> [2] Ma, T., Yao, J., He, D., Peng, S., Li, Y., Liu, S., & Tian, Z. (2025). Context-Aware Hierarchical Learning: A Two-Step Paradigm towards Safer LLMs. Advances in Neural Information Processing Systems (NeurIPS 2025).
>
> [3] Shi, Z., Wan, G., Wang, H., Li, R., Huang, Z., Zhao, W., Xiao, Y., Luo, X., Yang, C., Sun, Y., & Wang, W. (2025). Don’t Forget the Enjoin: FocalLoRA for Instruction Hierarchical Alignment in Large Language Models. Advances in Neural Information Processing Systems (NeurIPS 2025).

---

> > ### Comment · Reviewer_Rtoh · 2025-11-24
> >
> > Thank the authors for their detailed explanation. I am satisfied with their response.

---

### Official Review · Reviewer_x1XQ · 2025-10-29

**Soundness:** 3
**Presentation:** 3
**Contribution:** 3
**Rating:** 4
**Confidence:** 4

**Summary:**

This paper proposes ASIDE (Architecturally Separated Instruction-Data Embeddings), a novel, parameter-free architectural modification to Large Language Models (LLMs) aimed at mitigating prompt injection vulnerabilities. The authors identify the lack of intrinsic separation between instructions (which should be executed) and data (which should be processed) as a root cause of these attacks.

ASIDE's core mechanism is to create distinct representations for instructions and data at the very first layer. It achieves this by applying a fixed, parameter-free 90-degree orthogonal rotation to the token embeddings of all inputs designated as "data," while leaving "instruction" token embeddings unchanged. This modified model is then instruction-tuned using a standard (non-adversarial) dataset.


The paper empirically demonstrates across a range of models (including Llama, Qwen, and Mistral) that ASIDE:

* Achieves substantially higher instruction-data separation (measured by the SEP score).

* Maintains model utility and performance, comparable to standard fine-tuned models (measured by SEP Utility and Alpaca Eval 1.0).

* Improves robustness against both indirect and direct prompt injection benchmarks, even without any dedicated safety training.

The authors supplement these findings with a strong set of interpretability analyses, including linear probing and causal interventions, to validate that the architectural separation persists through the model's layers and is causally linked to the improved safety.

**Strengths:**

* Novelty and Elegance: The proposed method is simple, highly novel, and elegant. Using a fixed, parameter-free orthogonal rotation is a clever architectural solution that avoids the overhead of additional parameters or complex, learnable components.



* Strong Empirical Validation: The claims are convincingly supported by experiments across a wide and diverse set of modern LLMs (Llama 2 7B/13B, Llama 3.1 8B, Qwen 2.5 7B, Qwen3 8B, and Mistral 7B). The improvement in instruction-data separation (SEP score, Figure 2a) is consistent and significant over all baselines.




* Practical Safety Improvement: A key contribution is that ASIDE provides a measurable and "free" improvement in robustness, particularly against indirect prompt injections (Table 1), without requiring any adversarial data or specialized safety fine-tuning. This makes it a highly practical method for improving the baseline safety of LLMs.


* Excellent Interpretability and Analysis: This is a standout strength of the paper. The authors provide deep insights into why ASIDE works.

* Clarity: The paper is exceptionally well-written, with clear illustrations (especially Figure 1) and a logical flow that makes the method and its evaluation easy to follow.

**Weaknesses:**

*Scope Limited to Pre-Defined Roles: The primary limitation is that ASIDE requires the functional role (instruction vs. data) of input tokens to be specified a priori by the system implementer. This is a reasonable assumption for many integrated applications (e.g., RAG, email clients). Still, it is not directly applicable to general-purpose, multi-turn chatbots where a single user turn might contain a mix of data (e.g., continuing a story) and new instructions (e.g., "now change the character's name"). The authors acknowledge this limitation.

* Limited Justification for Rotation Choice: The paper specifies a 90-degree ($\frac{\pi}{2}$) isoclinic rotation and justifies it based on computational efficiency (it simplifies to coordinate swapping and negation). While practical, this leaves the theoretical justification underexplored. The paper would be strengthened by an ablation study comparing this specific rotation to other angles (e.g., 45 degrees) or other types of parameter-free orthogonal transformations to show that 90 degrees is an optimal or robust choice.

**Questions:**

1. Multi-turn Scenarios: The paper's scope is explicitly limited to single-turn, system-level applications where roles are pre-defined. Could you elaborate on how you envision ASIDE being adapted for a multi-turn conversational setting? For instance, could a classifier be trained to assign "instruction" or "data" roles to spans of a user's turn before the embedding rotation is applied?



2. Choice of Rotation: Your justification for the 90-degree rotation is its computational efficiency. Did you experiment with other rotation angles (e.g., 45, 180 degrees) or other types of parameter-free orthogonal transformations? How sensitive is the model's SEP score and ASR to this specific choice?

3. ASIDE vs. ISE Mechanism: The finding in Figure 3 that ISE's linear separability degrades in deeper layers while ASIDE's does not is a key differentiator. What is your hypothesis for this? Why do you believe the network can "undo" or "ignore" a learnable offset (ISE) more easily than it can a fixed rotation (ASIDE)?


4. Direct Injection Nuance: The robustness gains on direct prompt injection (Table 1) appear less pronounced than on indirect injection, with negligible improvement on the RuLES benchmark. Does this suggest ASIDE is primarily effective against attacks based on role confusion (data-as-instruction), rather than attacks that try to override a known instruction (jailbreaks)?

---

> ### Author Response · Authors · 2025-11-20
> **Official Comment by Authors: Part 1/2**
>
> Dear Reviewer x1XQ,
>
> We thank you for your review and positive assessment of our work. We are pleased that you appreciated the “novelty and elegance” of our approach, found our validation to be “strong,” and considered our interpretability analysis “excellent.” We are happy to address the concerns you raised below
>
> > Choice of Rotation: Limited Justification for Rotation Choice: The paper specifies a 90-degree  isoclinic rotation and justifies it based on computational efficiency (it simplifies to coordinate swapping and negation). While practical, this leaves the theoretical justification underexplored. The paper would be strengthened by an ablation study comparing this specific rotation to other angles (e.g., 45 degrees) or other types of parameter-free orthogonal transformations to show that 90 degrees is an optimal or robust choice.
>
> Thank you for this comment! Based on your suggestion, we ran an additional ablation study and can confirm that the 90-degree rotation is optimal for maximizing the model’s separation.
>
> We conducted an ablation on ASIDE for Qwen3-8B, training with different angles $\theta \in \{\pi/8, \pi/4, 3\pi/8, \pi/2, 5\pi/8, 3\pi/4, 7\pi/8, \pi\}$. We observe a clear trend: our separation metric (SEP) increases monotonically as $\theta$ moves from $0$ to $\pi/2$, and then decreases for larger angles. The 90-degree rotation is the most extreme rotation we can apply to differentiate embeddings, and our experiments show it is the most effective.
>
> We have updated Appendix A.1 in the revised paper to illustrate SEP vs. Angle. For your convenience, we provide a summary of the results in the table below:
>
>
> | Rotation Angle (θ) | π/8 | π/4 | 3π/8 | **π/2** | 5π/8 | 3π/4 | 7π/8 | π |
> |-------------------|-----|-----|------|---------|------|------|------|---|
> | SEP Score | 44.4 ± 0.7 % | 45.1 ± 0.7 % | 52.8 ± 0.6 % | **71.4 ± 0.6 %** | 40.2 ± 0.6 % | 41.7 ± 0.6 % | 52.8 ± 0.6 % | 51.3 ± 0.6 % |
>
>
>
> > Multi-turn Scenarios: The paper's scope is explicitly limited to single-turn, system-level applications where roles are pre-defined. Could you elaborate on how you envision ASIDE being adapted for a multi-turn conversational setting? For instance, could a classifier be trained to assign "instruction" or "data" roles to spans of a user's turn before the embedding rotation is applied?
>
> We wish to emphasize that the current scope of ASIDE is intentionally designed for integrated applications (e.g., email clients) or specific safety scenarios with clear instruction-data separation, such as AI Control or Oversight. In these settings, where models serve as monitors, it is critical that their predefined behavior cannot be hijacked by untrusted data inputs.
>
> In a multi-turn conversation setting, assigning roles via a classifier wouldn’t be necessary. In most realistic threat models, attacks come from the untrusted external content (tool outputs, web search results, uploaded files, etc). This external content can be labeled as data, while user input would serve as an instruction.
>
> ... (to be continued)

---

> > ### Author Response · Authors · 2025-11-20
> > **Official Comment by Authors: Part 2/2**
> >
> > > ASIDE vs. ISE Mechanism: The finding in Figure 3 that ISE's linear separability degrades in deeper layers while ASIDE's does not is a key differentiator. What is your hypothesis for this? Why do you believe the network can "undo" or "ignore" a learnable offset (ISE) more easily than it can a fixed rotation (ASIDE)?
> >
> > We believe that an orthogonal rotation (ASIDE) is a much more persistent change to the embedding space than a learnable offset (ISE). In ISE, the role information is encoded as an additive embedding. This signal lives in a low-rank subspace and is easiest for later layers to ignore, e.g., through applying subsequent LayerNorm and linear projections. This is consistent with Fig. 3: ISE’s linear separability curve merges with the Base/Vanilla curves after about 8 layers, indicating that its effect is limited to early layers. In contrast, ASIDE enforces an orthogonal rotation of instruction vs. data embeddings. “Undoing” it would require systematically learning an inverse rotation, but only conditionally for specific tokens. Consequently, ASIDE gives stronger linear separability in later layers, which we expect to matter more for high-level decisions such as “execute vs. translate this sentence.”
> >
> > > Direct Injection Nuance: The robustness gains on direct prompt injection (Table 1) appear less pronounced than on indirect injection, with negligible improvement on the RuLES benchmark. Does this suggest ASIDE is primarily effective against attacks based on role confusion (data-as-instruction), rather than attacks that try to override a known instruction (jailbreaks)?
> >
> > This interpretation is correct. ASIDE is specifically designed to make the boundary between instructions and data clear and explicit.
> >
> > - - -
> >
> >
> >  We thank you for your thoughtful and encouraging review and your helpful suggestions. We believe the additional ablation study makes our method more grounded and explicitly justifies our design choices. Given your overall positive assessment of our work, we were surprised by the current rating. We hope that our new experiments and clarifications address your concerns and that you are satisfied with the revisions. If this is the case, we kindly ask you to consider raising the score. Otherwise, we are happy to discuss any remaining questions.

---

> ### Comment · Reviewer_x1XQ · 2025-11-21
>
> Dear Authors,
>
> Thank you for the detailed responses to my concerns. After carefully reviewing your feedback and the paper again, I have decided to raise my score.

---

### Official Review · Reviewer_YbgE · 2025-10-31

**Soundness:** 2
**Presentation:** 2
**Contribution:** 2
**Rating:** 2
**Confidence:** 4

**Summary:**

This paper proposes ASIDE, a lightweight architectural modification that enforces structural separation between instruction and data embeddings in large language models. By applying orthogonal transformations to distinguish instruction and data tokens, ASIDE improves robustness against prompt injection attacks without introducing extra parameters. Experiments show significant gains in instruction–data separation and lower attack success rates.

**Strengths:**

1. The proposed method is simple, easy to understand, and conceptually interesting.

2. The proposed method is computationally efficient and does not introduce any additional parameters.

3. The paper includes insightful analyses explaining why ASIDE achieves better instruction–data separation performance.

**Weaknesses:**

1. As stated in the paper, the proposed method is designed as a defense against prompt injection attacks. Therefore, I believe the paper should include comparisons with the most recent fine-tuning-based defenses such as StruQ [1], SecAlign [2], and Meta-SecAlign [3]. The current baselines used for comparison appear relatively weak.

2. Although the ASR of ASIDE in Table 1 is lower than that of other baselines, some numbers remain quite high (exceeding 60%), suggesting that ASIDE may not be a very effective defense against prompt injection in practice.

3. I find the utility results not very convincing. Specifically, the results on AlpacaEval may be over-estimated, since the training set (Alpaca-Clean) is very similar to AlpacaEval, potentially leading to distribution overlap. Moreover, Qwen-based models (which already possess strong instruction-following capabilities) perform well on SEP but worse than other methods on AlpacaEval, indicating possible evaluation bias.

4. I think the authors’ statement “we use plain pretrained models rather than instruction- or safety-tuned models to avoid biasing the safety evaluations” somewhat unconvincing. As we know, the utility of the pretrained Llama models is very poor (also as shown in Figure 2), and such models generally lack instruction-following capacity. In realistic deployments, using instruction-tuned models is unavoidable. Therefore, the paper should additionally evaluate ASIDE on instruction-tuned versions of those models, and compare their utility with the original instruction-tuned baselines, or at least report the results of the instruction-tuned models. This would better demonstrate the compatibility between ASIDE and instruction tuning.

[1].Chen, Sizhe, et al. "{StruQ}: Defending against prompt injection with structured queries."

[2].Chen, Sizhe, et al. "Secalign: Defending against prompt injection with preference optimization."

[3].Chen, Sizhe, et al. "Meta SecAlign: A Secure Foundation LLM Against Prompt Injection Attacks."

**Questions:**

Please see the weakness part above.

---

> ### Author Response · Authors · 2025-11-20
> **Official Comment by Authors: Part 1/2**
>
> Dear Reviewer YbgE,
>
> We thank you for your review and for constructive feedback!
>
> We agree that any dedicated defense requires a fair comparison to existing approaches and necessitates adaptive attacks. However, **we do not propose ASIDE as a defense, but rather as a principled mechanism for separating instructions from data.**
>
> Our central point is that even during completely benign instruction-tuning, it is possible to achieve significant improvements in instruction-data separation, and consequently, some “free” gains in model safety, without any dedicated safety interventions. We demonstrate that such results can be achieved with a simple architectural modification.
>
> > As stated in the paper, the proposed method is designed as a defense against prompt injection attacks. Therefore, I believe the paper should include comparisons with the most recent fine-tuning-based defenses such as StruQ, SecAlign, and Meta-SecAlign. The current baselines used for comparison appear relatively weak.
>
> We would like to emphasize that we explicitly refrain from framing ASIDE as a defense. ASIDE is a method for improving instruction-data separation of LLMs. We show that improving this separation naturally leads to a boost in safety, a valuable **byproduct** achieved without any dedicated safety interventions.
>
> We acknowledge that on specific attack benchmarks, defenses like StruQ or SecAlign may outperform ASIDE because they boil down to adversarial training explicitly tailored to be robust to prompt injections. However, as shown by recent work [1], these defenses are prone to stronger adaptive attacks, which we believe is because neither of them addresses the **core tenet** of the problem.
>
> This is precisely why we consider methods like ASIDE to be valuable for the community. From a system design perspective, there is a clear distinction between instruction and data in many downstream applications. We believe that such a distinction can and should be embedded directly in the model.
>
>
> > Although the ASR of ASIDE in Table 1 is lower than that of other baselines, some numbers remain quite high (exceeding 60%), suggesting that ASIDE may not be a very effective defense against prompt injection in practice.
>
> Indeed. Despite that, we can appreciate how it consistently decreases ASR in the majority of cases, even though the model was not trained on any adversarial examples, and the only difference with Vanilla is the 90-degree embedding rotation.
>
> > I find the utility results not very convincing. Specifically, the results on AlpacaEval may be over-estimated, since the training set (Alpaca-Clean) is very similar to AlpacaEval, potentially leading to distribution overlap. Moreover, Qwen-based models (which already possess strong instruction-following capabilities) perform well on SEP but worse than other methods on AlpacaEval, indicating possible evaluation bias.
>
> We want to emphasize that all methods were trained in the same way on Alpaca-Clean, meaning, ASIDE could not receive any biased advantages compared to baselines.
>
> We start with pre-trained models that didn’t undergo any instruction tuning, and therefore, they cannot possibly possess strong instruction-following capabilities. Also, ASIDE performs on par with Vanilla on Qwen-based models on AlpacaEval: +0.7% on Qwen 2.5 7B and -0.2% on Qwen 3 8B, both of which are within error bars. In this experiment, we care about our method having higher separation while not losing on performance (utility). While the second baseline (ISE) has higher utility compared to both Vanilla and ASIDE, this is inconsequential, as having higher utility but no improvement to separation is not the goal of this line of research.
>
> - - -
>
> References:
>
> [1] Nasr, Milad, et al. "The attacker moves second: Stronger adaptive attacks bypass defenses against llm jailbreaks and prompt injections." arXiv preprint arXiv:2510.09023 (2025).

---

> > ### Author Response · Authors · 2025-11-20
> > **Official Comment by Authors: Part 2/2**
> >
> > > I think the authors’ statement “we use plain pretrained models rather than instruction- or safety-tuned models to avoid biasing the safety evaluations” somewhat unconvincing. As we know, the utility of the pretrained Llama models is very poor (also as shown in Figure 2), and such models generally lack instruction-following capacity. In realistic deployments, using instruction-tuned models is unavoidable. Therefore, the paper should additionally evaluate ASIDE on instruction-tuned versions of those models, and compare their utility with the original instruction-tuned baselines, or at least report the results of the instruction-tuned models. This would better demonstrate the compatibility between ASIDE and instruction tuning.
> >
> > This appears to be a misunderstanding. We do evaluate on instruction-tuned models, just not on models downloaded in this form from the web, but by performing the instruction-tuning ourselves. This is necessary for the sake of a fair comparison: for ASIDE (and ISE) we need to perform fine-tuning after the architectural change, to recover after the architectural changes. We cannot follow exactly the same steps as instruction-tuned models from the web, as the data used and the exact procedure is not made public. Consequently, the results for ASIDE/ISE would not be comparable with public instruction-tuned models. To make them comparable, we perform the instruction-tuning for those models ourselves as well (the Vanilla baseline). As you also observed, the instruction-tuning does have a clear impact on instruction-following, as the  difference to the Base baseline shows.
> >
> > While we do agree that for a developer seeking to apply open-source LLMs to their applications “using instruction-tuned models is unavoidable”, we are targeting a real-world setting in which the model creators themselves adopt ASIDE inside of their instruction-tuning pipeline.
> >
> > - - -
> >
> >
> > We hope our clarifications have helped to better define the scope and merit of our method. We remain open to further discussion should you have additional questions. If our response has addressed your concerns, we would kindly ask you to consider raising your score.

---

> > > ### Author Response · Authors · 2025-11-26
> > > **We would appreciate an opportunity to engage with you**
> > >
> > > Dear Reviewer YbgE,
> > >
> > > As the discussion period draws to its end, we would greatly appreciate the opportunity to engage with you regarding our rebuttal.
> > >
> > > We hope that our clarifications regarding the scope of the paper, its non-trivial safety improvements, and analysis we provide in the paper have at least partially addressed your concerns. We believe ASIDE contributes a complementary perspective to the prompt injection defense landscape, and we would value your feedback on whether our responses have helped clarify the paper's contributions and limitations.
> > >
> > > We have already addressed concerns of all other reviewers and as the current score puts the work at the borderline acceptance, we would greatly appreciate the opportunity to further discuss the scope of our paper and would be happy to address any additional questions you might have.
> > >
> > > Best regards,
> > > The Authors

---

> > > > ### Comment · Reviewer_YbgE · 2025-11-27
> > > >
> > > > Thank you for the detailed response. I believe that my main concerns have been largely addressed. I agree that describing ASIDE as a “method for improving instruction–data separation” is indeed more appropriate. The key contribution of the paper is showing that a very simple structural modification to the model can improve LLM safety while incurring almost no utility degradation.
> > > >
> > > > However, I am still a bit confused about the distinction the authors draw between ASIDE and a defense. In practice, the effect appears quite similar to that of a defense: defenses aim to reduce ASR while minimally affecting utility, which is precisely what ASIDE achieves. Moreover, instruction–data separation seems to matter primarily in the context of safety. For that reason, I do not see an issue with characterizing ASIDE as a defense, at least from a functional perspective.
> > > >
> > > > Given the clarifications provided, I am willing to raise my score to a 4, and I would be open to increasing it further if the authors (i) include comparisons with recent prompt-injection defenses to demonstrate ASIDE’s strengths, or (ii) provide a clearer explanation of how their method conceptually differs from a defense.

---

> > > > > ### Author Response · Authors · 2025-12-02
> > > > >
> > > > > Dear Reviewer YbgE,
> > > > >
> > > > > Thank you for engaging in the discussion and **raising your original score from 2 to 4.**
> > > > >
> > > > > We agree that from a practical standpoint the endgoal of instruction-data separation is model safety. However, architectural separation of instructions and data approaches model safety from a different angle than typical data-centric defenses such as StruQ and SecAlign, and aim to reach this goal through a different mechanism.
> > > > >
> > > > > To support this point, we ran additional experiments evaluating the Meta-SecAlign-8B in our safety evaluations. Note that Meta-SecAlign-8B is an adversarially trained from stronger baseline of meta-llama/Llama-3.1-8B-Instruct, while ASIDE is trained from meta-llama/Llama-3.1-8B the base model. Thus ASIDE involves only improvements coming from the architectural separation, while the SecAlign represents the data-centric approach to the model safety problem.
> > > > > | Model                          |             |         |        |       | Dataset    |            |          |           |
> > > > > | ------------------------------ | ----------- | ------- | ------ | ----- | ---------- | ---------- | -------- | --------- |
> > > > > |                                | TensorTrust | Gandalf | Purple | RuLES | BIPIA-text | BIPIA-code | StruQ-ID | StruQ-OOD |
> > > > > | ASIDE Llama 3.1 8B             | 36.6        | 50.5    | 79.9   | 78.4  | 4.1        | 9.2        | 41.3     | 47.3      |
> > > > > | SecAlign Llama 3.1 8B Instruct | 54.7        | 12.6    | 97.9   | 41.6  | 1.12       | 39.6       | 5.4      | 12.2      |
> > > > >
> > > > > ASIDE outperforms SecAlign on TensorTrust, Purple and BIPIA-code, while on other benchmarks, SecAlign performs better in terms of model safety. SecAlign is trained on a Prompt-Injected Alpaca dataset and performs well on the StruQ benchmark, which is also based on malicious instructions injected into the Alpaca dataset. However, on dissimilar datasets, such as TensorTrust or BIPIA-code, the performance suffers, indicating a generalization gap. ASIDE, in contrast, does not involve training on any adversarial examples and provides consistent improvements over the base model across 7/8 safety benchmarks
> > > > >
> > > > > This result underscores that architectural approaches to LLM security are a promising and underexplored approach, orthogonal to data-centric methods, with its own, decorrelated strengths and weaknesses. This paves a clear way forward toward a “swiss-cheese” model of LLM security, combining multiple approaches and reducing overall risks.

---

### Meta-Review · Area_Chair_q2re · 2026-01-05

**Summary:**

This paper proposes a novel method to improve the separation of instructions and data in language models, which has direct applications to enhancing the robustness to prompt injection. All reviewers find the studied setting novel, and the results provide new insights. The authors’ rebuttal has successfully addressed the major concerns of reviewers, primarily centered on the positioning of this work and the robustness of the proposed method. All reviewers are quite positive after the authors' responses, and I recommend accetance.

**Reviewer Concerns:**

I believe all concerns were properly addressed.

**Reviewer Scores:**

All reviewers are willing to increases the scores, per the discussion.

---

### Decision · Program_Chairs · 2026-01-26

Accept (Poster)